# Combined Immune Checkpoint Blockade and Helixor^®^ Therapy in Oncology: Real-World Tolerability and Subgroup Survival (ESMO GROW)

**DOI:** 10.3390/ijms26083669

**Published:** 2025-04-12

**Authors:** Anja Thronicke, Patricia Grabowski, Juliane Roos, Hannah Wüstefeld, Christian Grah, Sophia Johnson, Friedemann Schad

**Affiliations:** 1Network Oncology Registry, Research Institute Havelhöhe, Kladower Damm 221, 14089 Berlin, Germany; sophia.johnson@havelhoehe.de; 2Interdisciplinary Oncological Centre, Hospital Havelhöhe, Kladower Damm 221, 14089 Berlin, Germany; patricia.grabowski@havelhoehe.de (P.G.); juliane.roos@havelhoehe.de (J.R.); 3Charité Fatigue Centrum, Charité–Universitätsmedizin Berlin, Augustenburger Platz 1, 13353 Berlin, Germany; 4Lung Cancer Center, Hospital Havelhöhe, Kladower Damm 221, 14089 Berlin, Germany; hannah.wuestefeld@havelhoehe.de (H.W.); christian.grah@havelhoehe.de (C.G.)

**Keywords:** PD-1 inhibitor, PD-L1 inhibitor, survival, Helixor^®^ *Viscum album* therapy, non-small cell lung cancer, lung cancer

## Abstract

Real-world data (RWD) play a crucial role in identifying key subgroups and assessing multimodal oncology therapies, including integrative and palliative care. Immune checkpoint blockade (ICB) has improved survival, and its combination with complementary therapies like *Viscum album* L. extracts (VA) may enhance outcomes. This RWD study, based on the Network Oncology registry and ESMO-GROW guidelines, analyzed oncological patients receiving PD-1/PD-L1 inhibitors alone or with Helixor^®^ VA (HVA) extracts. Primary and secondary objectives were tolerability and overall survival. Statistical analyses included Kaplan–Meier survival curves and Cox regression. Among 405 cancer patients, 344 received ICB alone (CTRL) and 61 received ICB + HVA (COMB). Lung cancer was predominant (78.6%). Adverse event-related discontinuation was lower in COMB (4.9% vs. 6.4%, *p* = 0.25). In non-small cell lung cancer (NSCLC) patients, the 3-year survival rate was significantly higher in COMB (34.3% vs. 17.2%, *p* = 0.02). In female NSCLC patients, COMB was significantly associated with a reduced death risk of 91.2% (aHR: 0.088; 95% CI: 0.009–0.783). Our RWD findings show the favorable tolerability of combinatorial ICB + HVA in several tumor entities and underscore its potential to improve survival in NSCLC particularly in female NSCLC patients, warranting further investigation.

## 1. Introduction

The emergence of immune checkpoint blockade (ICB) therapies has transformed the way how cancer is treated nowadays, providing remarkable survival benefits across multiple malignancies [1]. ICB therapies enhance the immune system’s ability to fight cancer by targeting immune-regulatory pathways, including programmed cell death protein-1 (PD-1), programmed death-ligand 1 (PD-L1), and cytotoxic T-lymphocyte-associated protein 4 (CTLA-4) [1]. Innovations in ICB therapies are improving effectiveness across cancer types [2,3]. Continued research into optimizing ICB and exploring new combinations is key to enhancing cancer immunotherapy outcomes. Recent evidence suggests that incorporating European mistletoe *Viscum album* L. (VA) extracts into PD-1/PD-L1 therapy may improve overall survival in patients with advanced or metastatic non-small cell lung cancer (NSCLC) [4,5]. National guidelines on complementary oncology treatments also mention the use of VA alongside these immunotherapies [6]. Safety assessments indicate that using VA as an add-on to PD-1/PD-L1 inhibitors does not introduce safety risks [7,8,9]. Helixor^®^ *Viscum album* (HVA), a mistletoe extract widely used in complementary oncology, has garnered attention for its potential immunomodulatory and anti-cancer properties. Findings suggest the potential of VA therapy in improving patient prognosis across various settings [10,11,12,13]. However, outcomes may be different between tumor types, i.e., immune modulators such as PD-1 inhibitors or VA may be more effective in cancers like NSCLC because these tumors often have a higher mutational burden, producing more neo-antigens that can trigger an immune response [14]. In contrast, pancreatic cancer tends to have a lower mutational burden and a dense, immunosuppressive tumor microenvironment (TME), which may limit effectiveness of PD-1 inhibitors [15] and VA [16]. Preclinical and clinical studies suggest that HVA can stimulate the immune system [17], improve quality of life [10,18,19], and enhance tolerability of standard cancer therapies [5,18,20,21,22]. Nonetheless, the integration of HVA with ICB therapies remains underexplored, and clinical evidence is limited.

In recent years, real-world data (RWD) derived from clinical registries have become increasingly important for generating evidence in clinical studies in addition to randomized controlled trials (RCTs) [23,24,25]. RWD is increasingly recognized to be especially useful for identifying and integrating key subgroups being often excluded by RCTs such as elderly oncological patients, rare cancer types, and those groups with deteriorated performance status or with multiple comorbidities. In addition, RWD may be used to analyze the effect of multimodal therapies, for example, in integrative oncology [26]. Consequently, RWD studies provide critical insights into the effectiveness, safety, and applicability of therapies in broader, unselected populations [24]. RWD studies also allow for the analysis of treatment patterns, adverse events, and survival outcomes in heterogeneous patient cohorts over extended periods [27]. These studies can examine how factors such as tumor stage, treatment sequencing, and adjunctive therapies influence outcomes in diverse clinical settings [28]. In addition, registry-based RWD studies are valuable for assessing complementary and integrative oncology interventions like HVA, as such therapies are often underrepresented in traditional clinical trials.

Our study utilized real-world registry data to evaluate the safety of combining ICB with HVA therapy in a cohort of 405 oncological patients and the efficacy of this combination in a subgroup of 312 patients with non-small cell lung cancer (NSCLC). This registry-based approach enabled the inclusion of a diverse patient population, encompassing various cancer types and treatment regimens, reflecting routine clinical practice.

## 2. Results

### 2.1. Baseline Characteristics

In total, four hundred and five (*n* = 405) oncological patients being treated with ICB were included in the study. Of them, 344 received ICB only (CTRL) and 61 received ICB with additional Helixor^®^ VA therapy (COMB) (Figure 1). For 405 patients, safety analyses were performed. Out of these 405 participants, 312 patients with NSCLC survival data were available.

The median age of the total cohort was 66 years (interquartile range 59–74). Participants from the COMB group were in median four years younger than participants from the CTRL group; the difference was significant, see Table 1. Almost half of the total cohort was female (47.9%); no significant differences in gender were observed in both groups. The most common cancer type receiving ICB was bronchus and lung cancer (78.8%) followed by breast cancer (7.1%), melanoma (3.2%), and kidney cancer (2.5%). Other cancer entities included were urinary cancer, mesothelioma, bladder cancer, esophagus cancer, colon cancer, liver cancer, cervix uteri cancer, Hodgkin lymphoma, bile duct cancer, laryngeal cancer, stomach cancer, anal cancer, parotid gland cancer, tonsil cancer, and endometrium cancer. These other cancer entities constituted around 6% of all entities. Fewer patients in the COMB group had lung cancer (55.7%) than in the CTRL group (82.8%) but a higher proportion of breast cancer patients were observed in the COMB group (23%) vs. the CTRL group (4.4%), see Table 1.

### 2.2. Anti-Neoplastic Treatment

Chemotherapy was applied in the majority of patients (89.1%) followed by PD-1 inhibitors (71.1%), radiation (51.1%), PD-L1 inhibitors (25.4%), and surgery (24.2%), see Table 2. A higher proportion of patients in the CTRL group (64.8%) received first-line ICB compared to COMB (42.6%,), and the difference was significant. Here, the COMB group received 20.3% less first-line PD-1 inhibitor therapy compared to CTRL. No significant difference between both groups was observed with regards to PD-1, PD-L1, or CTL-A4 inhibitor treatment. As for ICB, PD-1 inhibitors were the most applied group as mentioned above, followed by PD-L1 inhibitors (25.4%) and CTL-4A inhibitors (0.7%). Within the group of PD-1 inhibitors, pembrolizumab was applied to the highest patient group (CTRL: 59.6%; COMB: 47.5%) followed by nivolumab (CTRL: 13.4%; COMB: 16.4%) and spartalizumab (CTRL: 0.6%; COMB: 0), see Appendix A. As for PD-L1 inhibitors, atezolizumab was the most common inhibitor (CTRL: 18.6%; COMB: 23.0%) followed by durvalumab (CTRL: 5.5%; COMB: 8.2%) and avelumab (CTRL: 0.3%; COMB: 0%), see Appendix A. Last but not least, CTLA-4 inhibitor therapy ipilimumab was applied in 2% of CTRL patients and in combination with nivolumab in 0.3% of CTRL patients, as shown in Appendix A.

### 2.3. Antineoplastic Treatment in Patients with Non-Small Cell Lung (NSCLC) Cancer

In the subgroup of NSCLC patients, the treatment regime revealed as well that chemotherapeutic agents were applied to the majority of patients (95.5) followed by PD-1 inhibitors (72.4%), radiation (54.8%), PD-L1 inhibitors (27.9%), and surgery (20.8%), see Table 3. A significantly higher proportion of patients in the CTRL group (71.6%) received first-line immune checkpoint blockade (ICB) therapy compared to the COMB group (52.9%), representing a 17.7% absolute difference between the groups, *p* = 0.03. As for PD-1, PD-L1, or CTL-4A inhibitor application, there were no significant differences between both groups. NSCLC patients from the COMB group received only PD-1 and PD-L1 inhibitors. Ten patients had early stage I-II NSCLC in the CTRL group and one patient in the COMB group, while 271 patients had late-stage III-IV NSCLC and 33 in the COMB group. The difference was not significant (*p* = 0.41).

### 2.4. Antineoplastic Treatment in Female Patients with NSCLC

Female NSCLC patients were primarily treated with chemotherapy (94.2%), PD-1 inhibitors (71.9%), radiation (54%), and surgery (19.4%), see Table 4. While there was no significant difference in the use of first-line ICB, 24.4% of female NSCLC patients in the CTRL group received first-line ICB compared to those in the COMB group.

No significant differences were observed between the two groups regarding PD-1, PD-L1, or CTLA-4 inhibitor therapy. Female NSCLC patients in the COMB group received only PD-1 and PD-L1 inhibitors, whereas those in the CTRL group also received CTLA-4 inhibitor therapy.

Regarding disease stage, two patients in the CTRL group (1.6%) and one in the COMB group (8.3%) had early-stage (I-II) NSCLC, while 121 patients in the CTRL group (95.3%) and 11 in the COMB group (91.7%) had late-stage (III-IV) NSCLC. The difference was not statistically significant (*p* = 0.29). 

### 2.5. Add-On HVA Treatment

HVA therapy was applied in addition to ICB in the COMB group. Among the patients from the COMB group, the majority of patients received ICB in combination with intravenous Helixor^®^ P (37.7%), intravenous Helixor^®^ A (21.3%), or intravenous Helixor M (16.4%), see Table 3. A small proportion of patients receiving ICB with intravenous Helixor A also received subcutaneous Helixor^®^ A (8.2%), see Table 4. In addition, a small proportion of patients receiving ICB with intravenous Helixor^®^ M or Helixor^®^ P also received subcutaneous Helixor^®^ M (1.6%) or subcutenous Helixor^®^ P, respectively (1.6%), see Table 5.

### 2.6. Tolerability of ICB with and Without HVA

Due to ICB adverse events, therapy was discontinued in 25 out of 405 patients (6.2%) in the total cohort. In the CTRL group, 22 patients (6.4%) discontinued ICB therapy due to adverse events, whereas in the COMB group, only three patients (4.9%) did so, see Figure 2. The difference was not significant (*p* = 0.25). All patients from the COMB group received intravenous HVA.

### 2.7. Effectivenss of Immune Checkpoint Blockade with and Without HVA

#### 2.7.1. Effectiveness of ICB and HVA in a Subgroup of Patients with NSCLC

The Kaplan–Meier survival curve performed in 312 patients with NSCLC revealed a survival advantage for the COMB in comparison to the CTRL group, see Figure 3 and Table 6.

The 3-year survival rate was 34.3% for the COMB group, which was double that of the CTRL group (17.2%). The difference was statistically significant with *p* = 0.022. The 5-year survival rate was 22% in the COMB group, two and a half times higher than the 8.8% observed in the control group, *p* = 0.058.

#### 2.7.2. Effectiveness of ICB and HVA in a Subgroup of Female NSCLC Patients

Kaplan–Meier overall survival curve in the female lung cancer subgroup revealed a survival advantage for the COMB compared to the CTRL group, see Figure 4 and Table 7. (This effect was not observed in male NSCLC patients, as shown in Appendix A). The median survival in the female NSCLC COMB group was not reached (95%CI: 19.3 months–NA) and higher than in the CTRL group where the median OS was 15.9 months (95%CI: 11.9–23.6 months, *p* = 0.06), see Table 6. Thus, the restricted mean survival time (RMST) was calculated. At 3 years, female NSCLC patients in the CTRL group had an RMST of 12.5 months, while those in the COMB group achieved an RMST of 20.8 months. The difference of 8.3 months indicates a better outcome for patients in the COMB group compared to the CTRL group.

The 3-year survival rate was 66.8% in the COMB group, compared to 26.9% for the control group. The difference was significant (*p* = 0.0123). The 5-year survival rate in the COMB group was four times higher at 50.1%, compared to 12% in the control group, a difference, which was statistically significant, *p* = 0.0021.

#### 2.7.3. Hazard of Death, Subgroup Female NSCLC

The multivariate cox proportional hazard analysis model in a subgroup of 137 females with NSCLC revealed that the COMB therapy was associated with a significant hazard of death risk reduction by 91.2% compared to the CTRL therapy (aHR: 0.088, 95%CI: 0.009 to 0.783), see Figure 5 and Appendix A. This suggests a true hazard of death reduction between 99.1% and 21.7%.

This benefit was independent of the other covariates, i.e., its benefit was not confounded or diminished by factors like chemotherapy, radiation, or surgery. In addition, this effectiveness of combined therapy was consistent across age, first-line treatment, and tumor stage groups.

## 3. Discussion

The findings of this RWD study reveal that combining immune checkpoint blockade (ICB) with Helixor^®^ VA therapy (HVA) shows a favorable safety profile in oncological patients and is associated with an improved survival in female NSCLC patients.

The safety profile observed in this study, with fewer discontinuations of ICB therapy in the COMB group, aligns with the literature highlighting the tolerability of add-on VA therapy when used alongside conventional cancer treatments [29,30]. Reduced discontinuation rates for the PD-1 inhibitor pembrolizumab, in particular, suggest that HVA therapy might help manage adverse events, potentially improving adherence to immunotherapy. This aligns with findings from clinical and real-world data studies suggesting that VA therapy can reduce chemotherapy- and radiotherapy-induced toxicities [11,31], reduce antibody-related toxicities [32], and does not increase ICB-associated toxicities [6,7,8,9]. In addition, add-on VA helps to maintain adherence to standard oncological therapy [21].

Tumor representation differed between the two groups, with only 33% of tumor entities documented in both. This is because our data reflect tumor entities from the registry that received real-life ICB or combinational ICB/Helixor VA therapies. Additionally, our intention was to highlight gaps in the real-life utilization of ICB and combinational ICB/Helixor VA therapy, which could be addressed in the future. Furthermore, younger patients were more frequently represented in the COMB group, suggesting a trend toward younger patients opting for complementary therapies. This aligns with previous findings that younger patients tend to integrate these therapies while adhering to guideline-based standard oncological treatments [33]. Lastly, a higher percentage of NSCLC patients was observed in the CTRL group. Based on this, we chose to conduct further analyses on specific subgroups, focusing particularly on the NSCLC group, and later on female NSCLC patients. During survival analyses, we accounted for the above-mentioned baseline differences using an adjusted multivariate regression.

The findings of this RWD study suggest a positive association of the combinational ICB plus Helixor^®^ VA therapy with improved overall survival in females with NSCLC. The hazard rate and the restricted mean survival time-based measures were in agreement regarding the statistical significance of the effect. The adjusted survival regression analysis balancing confounding effects revealed a significant effect of the combinational therapy on the adjusted hazard of death in female lung cancer patients. This effect was **independent** of age, stage, treatment line, or non-ICB neoplastic treatment. Interestingly, the 5-year survival rate in these patients increased more than fourfold when Helixor^®^ VA was added to ICB. Recent RWD studies of our group have already demonstrated that combining immune checkpoint inhibitors with VA therapy was linked to enhanced overall survival outcomes in NSCLC [4,5]. The significant improvement of the adjusted hazard of death observed in our female lung cancer subgroup adds an interesting dimension to our analysis so far. This effect was not observed in our male lung cancer group. Several studies have suggested that gender plays a role in the response to immune checkpoint inhibitors [34,35,36]. For example, a study published in 2020 found gender differences in response to immune checkpoint inhibitors also in NSCLC patients [34]. A systematic review and meta-analysis on the other hand did not find a significant difference between male or female NSCLC patients when treated with PD-1/PD-L1 checkpoint inhibitors [35]. Another study published in 2022 in the journal *Frontiers in Immunology* observed different expression patterns of PD-1 between male and female gender in NSCLC [36]. Thus, scientific results are inconsistent so far and the reason behind the potential gender disparity in immune checkpoint blockade response is not yet fully understood. Several genes and molecular pathways have been implicated in gender-related differences in immune responses and PD-L1 expression such as estrogen-related, sex hormone-related, immune-related, oncogene, and tumor suppression-related molecular pathways [37,38]. Genetic alterations of these genes may influence the expression of PD-L1 and immune checkpoint blockade [37,39]. While gender-specific responses to both immunotherapy and complementary treatments have been explored in the literature [35,40], more evidence is needed to elucidate whether female patients derive greater benefit from combinational approaches involving VA therapy.

The significant reduction in the hazard of death observed for the combination of ICB and HVA reflects growing data supporting the role of *Viscum album* (VA) extracts as an adjunct to oncological standard therapies including immune checkpoint blockade (ICB [8,9]. Thus, it seems that growing clinical evidence—from systematic reviews and meta-analyses to clinical trials and real-world data—underscores the transformative impact of VA extracts on survival outcomes in standard oncological treatments [12,13,41,42].

The findings of the present study align with the current literature exploring the integration of complementary therapies such as VA in oncology to enhance treatment outcomes. Saha and colleagues indicated in 2016 that VA extracts exhibit promising immunomodulatory properties, particularly in their potential to reshape the tumor microenvironment (TME) [43]. VA extracts, in particular VA lectins, strongly and selectively activate dendritic cells (DCs) and promote tumor-specific IFN-γ-mediated T-cell (Th1) activation [43]. At the same time, VA extracts are able to suppress the COX-2/prostaglandin E2 mediated production of regulatory T cells, the latter being associated with tumor evasion and tumor immunity. In summary, VA extracts uniquely and favorably modify the TME by enhancing immune activation without exacerbating immunosuppressive pathways [43]. Immunogenic cell death triggered by these properties of add-on VA extracts may unmask the tumor cells and thus reduce their resistance to anti-neoplastic treatment such as ICB. In addition, anti-tumor γβ T cells playing a role in NSCLC are targets for both VA extracts and PD-1/PD-L1 inhibitors resulting in a further explanation for synergistic effects of ICB and add-on VA therapy [44,45,46,47]. In summary, the ability of VA extracts to reshape the TME and enhance immune responses may support its potential as a complementary therapy in oncology.

There may be concerns that patients receiving complementary therapies, such as *Viscum album* extracts alongside immune checkpoint blockade (ICB), do not fully represent the overall patient population. However, it is well recognized that cancer patients increasingly incorporate complementary therapies into their treatment regimen, with usage rates reaching up to 80%. A systematic review found that the prevalence of complementary and alternative medicine (CAM) among cancer patients worldwide ranges from 40% to 80% [48]. In Europe, usage varies by country but generally falls between 30% and 70% [49].

### Limitations and Strength

Our results are promising but should be interpreted with caution given the real-world evidence nature of the study. The lack of randomization could introduce biases, and differences in baseline characteristics between groups (e.g., younger median age in the COMB group, different tumor distributions, and stage) may partially confound the results. Potential biases in this study were mitigated through the use of multivariable logistic regression analyses to control for confounding factors. Furthermore, the study is limited by the small sample size of subgroups which may introduce selection bias. Additionally, as this study is based on real-world registry data, the dosage and adherence to ICB and/or *Viscum album* extracts or ICB therapy were not always strictly documented. While this reflects routine clinical practice, it may introduce variability in treatment comparability. However, this study provides the first evidence of effectiveness data, highlighting its importance. Another notable strength of this real-world data study is its ability to reflect the actual clinical use of ICB in combination with complementary therapies, such as Helixor^®^ *Viscum album*.

## 4. Materials and Methods

### 4.1. Study Design

This real-world data (RWD) analysis utilized information from the Network Oncology (NO) registry [50], an oncological registry accredited by the German Cancer Society. The study followed ESMO-GROW criteria for RWD research [51].

#### 4.1.1. Patient Enrollment and Treatment

Patients were enrolled between 30 June 2015, and 25 June 2024. They received immune checkpoint blockade (ICB) therapy with PD-1/PD-L1 inhibitors, either alone (CTRL group) or in combination with Helixor^®^ *Viscum album* (HVA) extracts (COMB group). Treatment decisions, including the administration of HVA, were made at the discretion of the treating physician, following the summary of product characteristics (SmPC). The rationale for adding HVA therapy was to was to improve health-related quality of life, alleviate symptoms related to cancer and its treatment, and potentially extend survival.

#### 4.1.2. Study Objectives

The study’s primary objective was to assess the safety of anti-PD-1/PD-L1 treatment with and without Helixor^®^ therapy. The secondary objective was to descriptively analyze overall survival in oncology patients, identifying factors associated with reduced mortality risk. The study was registered (DRKS00013335).

#### 4.1.3. Eligibility Criteria and Data Collection

The analysis included cancer patients who had received immune checkpoint inhibitor (ICI) therapy, with (COMB group) or without HVA treatment (CTRL group), and were registered in the NO database.

Eligible patients were 18 years or older, of any gender, and provided written informed consent. Data extracted from the NO registry included demographics, diagnosis, tumor stage, treatment details, survival outcomes, tumor board decisions, and last contact. Documentation of HVA therapy included start/end dates, dosage, mode of application, and host tree type.

#### 4.1.4. Follow-Up

Routine follow-up occurred six months post-diagnosis and annually thereafter. Loss to follow-up was defined as the absence of follow-up visits.

### 4.2. Interdisciplinary Team

The study’s multidisciplinary team brought together experts from various fields, including clinical practice, epidemiology, and biostatistics. This combination of expertise was essential to conducting a successful RWD study in accordance with the ESMO-GROW criteria [52]. The team’s close collaboration ensured that all elements of the study were thoroughly addressed.

### 4.3. Ethics Issues

The study followed the ethical principles set out in the Declaration of Helsinki. Prior to their participation, all patients provided written informed consent. Ethical approval was granted by the ethics committee of the Medical Association Berlin (Eth-27/10).

### 4.4. Classification of Groups

Based on their treatment, patients were assigned to one of two groups: (1) the CTRL group, which received PD-1/PD-L1 inhibitors without Helixor^®^ VA therapy, or (2) the COMB group, which received PD-1/PD-L1 inhibitors along with additional Helixor^®^ VA therapy. Group assignment was non-randomized and determined by the physician after providing the patient with detailed treatment options, allowing them to make an informed decision. The Helixor^®^ VA therapy included extracts from Helixor GmbH only (Helixor Heilmittel GmbH, Rosenfeld, Deutschland).

### 4.5. Determination of Sample Size

To determine the necessary sample size for a two-sided test with an 80% power and a significance level of 5% using an allocation ratio of 0.2 (CTRL) to 0.8 (COMB) and an effect size of 0.6 [17,35], a total of 219 patients would be required. This included 44 patients in the COMB and 175 patients in the CTRL group, in order to confirm a statistically significant treatment effect, as outlined by Schoenfeld et al. [52].

### 4.6. Statistical Methods

Continuous variables were summarized using the median and interquartile range (IQR), while categorical variables were reported as absolute and relative frequencies. Group comparisons were performed using the unpaired Student’s *t*-test for continuous variables and the chi-square analysis for categorial variables. Kaplan–Meier survival curves were generated for both groups (CTRL and COMB) with survival time calculated from the start of PD-1/PD-L1 inhibitor therapy until death or the last follow-up.

Multivariate Cox proportional hazard models were used to assess factors influencing survival, adjusting for age, gender, tumor stage, ECOG performance status, PD-L1 status, and oncological treatment. Proportional hazard assumptions were verified before analysis. Tolerability was assessed for adverse event-related discontinuations rates, with descriptive statistics, providing a comparison between treatment groups. All statistical analyses were exploratory and conducted using R software version 4.1.2 (1 November 2021) and R-Studio (version 2022.02.2) [53]. The R package ‘survival’ (version 3.5-5) [38], ‘prodlim’ (version 2019.11.13) [54], and package ‘survminer’ (version 0.4.9) [55] were used for survival and event history analyses. We conducted sensitivity analyses, including subgroup analyses, to verify the robustness of our results and to minimize biases.

### 4.7. Data Analysis

Data distributions were visually inspected, and skewness was evaluated arithmetically. Patients with missing data were excluded from the analysis. Patients who had not died by the time of the analysis were censored. A year was defined as 365.25 days, and a month at 365.25/12 days.

## 5. Conclusions

Our RWD registry study shows favorable safety profiles in oncological patients combining ICB with Helixor^®^ VA. In addition, it reveals an association between this combinatorial therapy and an improved survival in female patients with NSCLC.

The observed benefits of adding VA therapy to ICB align with the existing literature that supports the role of complementary therapies in enhancing cancer treatment outcomes. Notably, this study is the first to suggest a potential gender-specific survival benefit of ICB + HVA therapy, warranting further exploration in larger prospective trials.

## Figures and Tables

**Figure 1 ijms-26-03669-f001:**
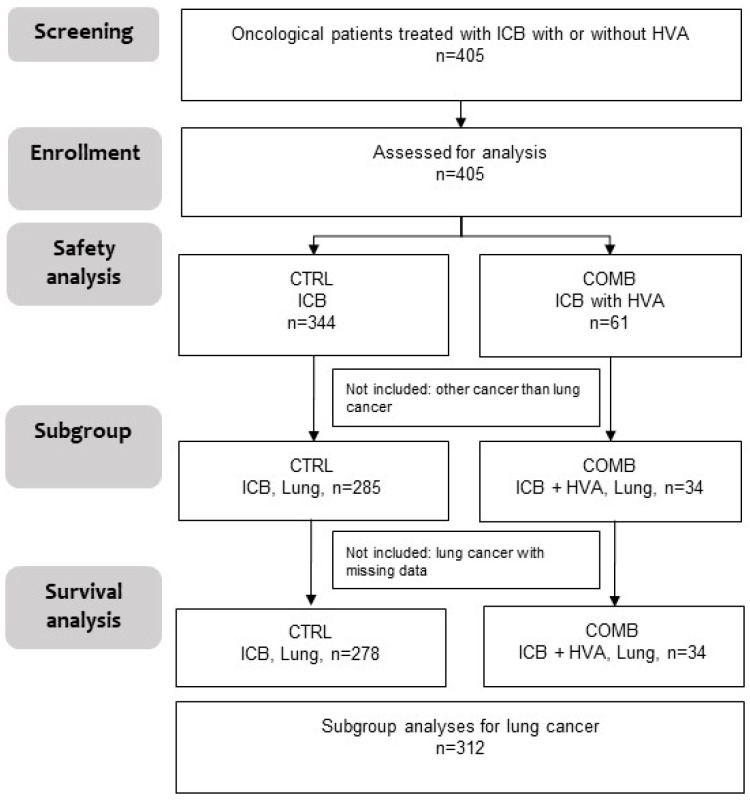
Study process flow. Oncological patients with ICB plus HVA or not, (*n* = 405), CTRL, ICB without HVA; COMB, ICB with HVA; ICB, immune checkpoint blockade; *n*, number; HVA, Helixor^®^ *Viscum album* therapy.

**Figure 2 ijms-26-03669-f002:**
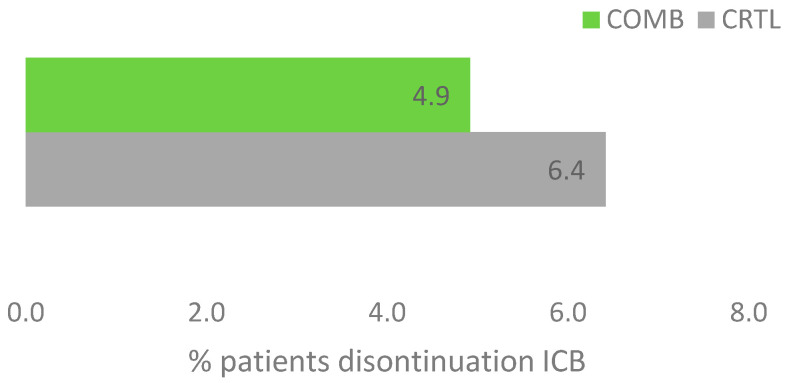
Discontinuation of ICB due to adverse events (*n* = 405), CTRL, ICB without HVA; COMB, ICB with HVA; ICB, immune checkpoint blockade; *n*, number; HVA, Helixor^®^ *Viscum album* therapy.

**Figure 3 ijms-26-03669-f003:**
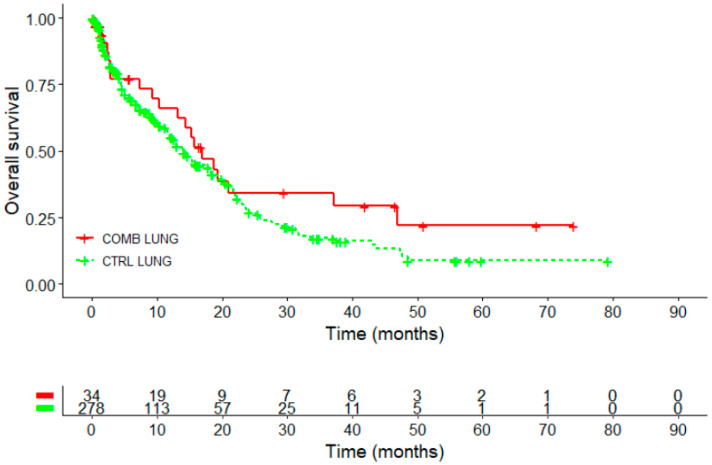
Kaplan–Meier survival analysis for overall survival in patients with NSCLC (*n* = 312); Log-rank test: X^2^ = 1.8, *p* = 0.2; CTRL, PD-1/PD-L1 inhibitors COMB, PD-1/PD-L1 inhibitors with HVA therapy; NSCLC, non-small cell lung cancer.

**Figure 4 ijms-26-03669-f004:**
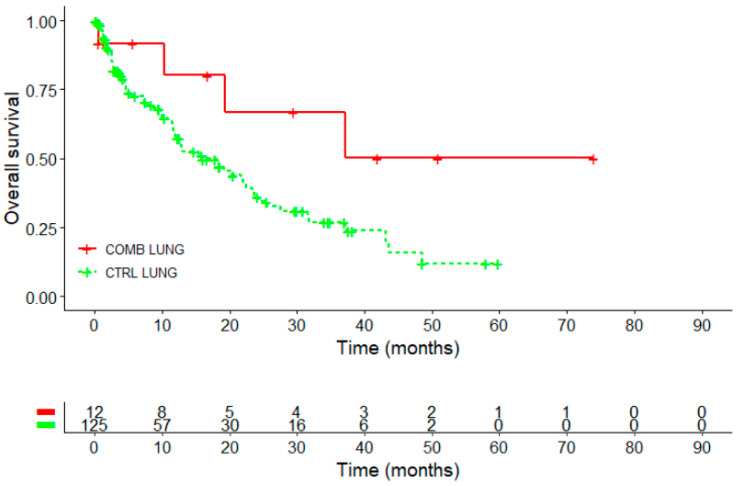
Kaplan–Meier survival analysis (overall survival) for female NSCLC patients treated with PD-1/PD-L1 inhibitors with or without Helixor^®^ VA (HVA) therapy (*n* = 137); Log-rank test: X^2^ = 3.6, *p* = 0.06; CTRL, PD-1/PD-L1 inhibitors COMB, PD-1/PD-L1 inhibitors with HVA therapy; *NSCLC, non-small cel7l lung cancer*.

**Figure 5 ijms-26-03669-f005:**
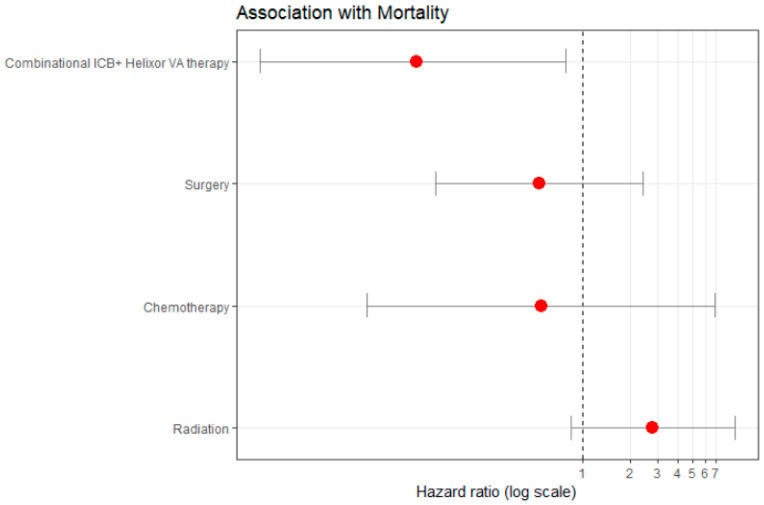
Cox proportional hazard analysis. Subgroup female patients with NSCLC (*n* = 137). Adjusted for standard oncological immune checkpoint therapy, stratified for age, tumor stage, and first-line therapy. Therapies with a hazard ratio less than 1 (to the left of 1) are considered to reduce mortality, while those with a hazard ratio greater than 1 (to the right of 1) are considered to increase mortality. The combinational ICB + HVA therapy reveals a significant reduction of hazard of death by 91.2%, *p* = 0.029 (aHR: 0.088, 95%CI: 0.009 to 0.783); aHR, adjusted hazard ratio; CI, confidence interval. ICB, immune checkpoint blockade; HVA, Helixor^®^ VA; NSCLC, non-small cell lung cancer.

**Table 1 ijms-26-03669-t001:** Characteristics of patients.

	Total Cohort (*n* = 405)	CTRL (*n* = 344)	COMB (*n* = 61)	*p*-Value
**Age at first diagnosis, median years (IQR)**	66 (59–74)	67 (60–75)	63 (56–69)	0.02
**Gender**				0.54
**Female, *n* (%)**	194 (47.9)	162 (47.1)	32 (52.5)	
**Male, *n* (%)**	210 (51.9)	181 (52.6)	29 (47.5)	
**Tumor type**				<0.001
**Bronchus and lung cancer, *n* (%)**	319 (78.8)	285 (82.8)	34 (55.7)	
**Breast cancer, *n* (%)**	29 (7.2)	15 (4.4)	14 (23.0)	
**Melanoma, *n* (%)**	13 (3.2)	9 (2.6)	4 (6.6)	
**Kidney cancer, *n* (%)**	10 (2.5)	10 (2.9)	0	
**Urinary cancer, *n* (%)**	5 (1.2)	4 (1.2)	1 (1.6)	
**Mesothelioma of pleura, *n* (%)**	4 (1.0)	4 (1.2)	0	
**Bladder cancer, *n* (%)**	4 (1.0)	3 (0.9)	1 (1.6)	
**Esophagus cancer, *n* (%)**	4 (1.0)	2 (0.6)	2 (3.3)	
**Colon cancer, *n* (%)**	3 (0.7)	3 (0.9)	0	
**Liver and intrahepatic bile duct cancer, *n* (%)**	2 (0.5)	1 (0.3)	1 (1.6)	
**Cervix uteri cancer, *n* (%)**	2 (0.5)	2 (0.6)	0	
**Hodgkin-lymphoma, *n* (%)**	1 (0.2)	0	1 (1.6)	
**Extrahepatic bile duct cancer, *n* (%)**	1 (0.2)	0	1 (1.6)	
**Laryngeal cancer, *n* (%)**	1 (0.2)	0	1 (1.6)	
**Stomach cancer, *n* (%)**	1 (0.2)	1 (0.3)	0	
**Anal cancer, *n* (%)**	1 (0.2)	0	1 (1.6)	
**Extrahepatic bile duct cancer, *n* (%)**	1 (0.2)	1 (0.3)	0	
**Malignant neoplasm of parotid gland, *n* (%)**	1 (0.2)	1 (0.3)	0	
**Malignant neoplasm of tonsil, *n* (%)**	1 (0.2)	1 (0.3)	0	
**Malignant neoplasm without specification of site, *n* (%)**	1 (0.2)	1 (0.3)	0	
**Endometrium cancer, *n* (%)**	1 (0.2)	1 (0.3)	0	
**Tumor stage according to UICC**				0.14
**Early stage, I + II, *n* (%)**	32 (7.9)	24 (7.0)	8 (13.1)	
**Advanced stage, III + IV, *n* (%)**	340 (84.0)	295 (85.8)	45 (73.8)	
**NA, *n* (%)**	32 (7.9)	24 (7.0)	8 (13.1)	

Patient characteristic, IQR—interquartile range; CTRL, ICB without HVA; COMB ICB with HVA; ICB, immune checkpoint blockade; *n*, number; HVA, Helixor^®^ *Viscum album* therapy.

**Table 2 ijms-26-03669-t002:** Characterization of antineoplastic therapy.

	Total Cohort (*n* = 405)	CTRL (*n* = 344)	COMB (*n* = 61)	*p*-Value
**Radiation, *n* (%)**	207 (51.1)	170 (49.4)	37 (60.7)	0.139
**Surgery, *n* (%)**	98 (24.2)	79 (23.0)	19 (31.1)	0.225
**Chemotherapy, *n* (%)**	361 (89.1)	311 (90.4)	50 (82.0)	0.08
**Hormone therapy, *n* (%)**	4 (1.0)	4 (1.2)	0	0.885
**PD-L1/PD-1/CTL-A4 inhibitors, *n* (%)**				0.189
**PD-L1 inhibitor, *n* (%)**	103 (25.4)	84 (24.4)	19 (31.1)	
**of these first-line PD-L1 inhibitor, *n* (%)**	70 (17.3)	60 (17.4)	10 (16.4)	
**PD-1 inhibitors, *n* (%)**	288 (71.1))	250 (72.7)	38 (62.3)	
**of these first-line PD-1 inhbitors, *n* (%)**	176 (43.5)	160 (46.5)	16 (26.2)	
**CTL-A4 inhibitors, *n* (%)**	3 (0.7)	2 (0.6)	2 (3.3)	
**of these first-line CTL-A4 inhibitors, *n* (%)**	0	0	0	
**PD-1/CTL-A4 inhibitors, *n* (%)**	10 (2.7)	8 (2.3)	2 (3.3)	
**of these first-line PD-1/CTL-A4 inhibitors, *n* (%)**	3 (0.7)	3 (0.9)	0	
**First-line ICB**	249 (61.5)	223 (64.8)	26 (42.6)	0.002

Neoplastic therapy including ICB therapy; *n*, number of patients; %, percent. CTRL, patients receiving ICB without HVA therapy; COMB, patients receiving ICB with VA therapy; CTL-A4, cytotoxic T-lymphocyte antigen 4; PD-L1, programmed death ligand 1; PD-1, programmed cell death protein-1; ICB, immune checkpoint blockade.

**Table 3 ijms-26-03669-t003:** Characterization of antineoplastic therapy in NSCLC patients.

	Total Cohort (*n* = 312)	CTRL (*n* = 285)	COMB (*n* = 34)	*p*-Value
**Radiation, *n* (%)**	171 (54.8)	148 (51.9)	23 (67.6)	0.12
**Surgery, *n* (%)**	65 (20.8)	56 (19.7))	9 (26.5)	0.48
**Chemotherapy, *n* (%)**	298 (95.5)	268 (94.0)	30 (88.2)	0.36
**PD-L1/PD-1/CTL-A4 inhibitors, *n* (%)**				0.52
**PD-L1 inhibitors, *n* (%)**	87 (27.9)	76 (26.7))	11 (32.4)	
**first-line PD-L1 inhibitors, *n* (%)**	66 (21.2)	58 (20.4)	8 (23.5)	
**PD-1 inhibitors, *n* (%)**	226 (72.4)	203 (71.2)	23 (67.6)	
**first-line PD-1 inhbitors, *n* (%)**	153 (49.0)	43 (50.2)	10 (29.4)	
**CTL-A4 inhibitors, *n* (%)**	1 (0.3)	1 (0.4)	0	
**first-line CTL-A4 inhibitors, *n* (%)**	0	0	0	
**PD-1/CTL-A4 inhibitor, *n* (%)**	5 (1.6)	5 (1.8)	0	
**first-line PD-1/CTL-A4 inhibitors, *n* (%)**	3 (1.0)	3 (1.1)	0	
**First-line ICB**	222 (71.2)	204 (71.6)	18 (52.9)	0.03

Neoplastic therapy including ICB therapy in NSCLC patients; *n*, number of patients; %, percent. CTRL, patients receiving ICB without HVA therapy; COMB, patients receiving ICB with VA therapy; CTL-A4, cytotoxic T-lymphocyte antigen 4; PD-L1, programmed death ligand 1; PD-1, programmed cell death protein-1; ICB, immune checkpoint blockade.

**Table 4 ijms-26-03669-t004:** Characterization of antineoplastic therapy in female NSCLC patients.

	Total Cohort (*n* = 139)	CTRL (*n* = 127)	COMB (*n* = 12)	*p*-Value
**Radiation, *n* (%)**	75 (54.0)	67 (52.8)	8 (66.7)	0.535
**Surgery, *n* (%)**	27 (19.4)	23 (18.1)	4 (33.3)	0.372
**Chemotherapy, *n* (%)**	131 (94.2)	120 (94.5)	11 (91.7)	1
**PD-L1/PD-1/CTL-A4 inhibitors, *n* (%)**				0.765
**PD-L1 inhibitors, *n* (%)**	38 (27.3)	35 (27.6)	3 (25.0)	
**first-line PD-L1 inhibitors, *n* (%)**	26 (18.7)	24 (18.9)	2 (16.7)	
**PD-1 inhibitors, *n* (%)**	100 (71.9)	91 (71.7)	9 (75.0)	
**first-line PD-1 inhibitors, *n* (%)**	63 (45.3)	60 (47.2)	3 (25.0)	
**CTL-A4 inhibitors, *n* (%)**	1 (0.7)	1 (0.8)	0	
**first-line CTL-A4 inhibitors, *n* (%)**	0	0	0	
**PD-1/CTL-A4 inhibitor, *n* (%)**	0	0	0	
**first-line PD-1/CTL-A4 inhibitors, *n* (%)**	0	0	0	
**First-line ICB**	89 (64.0)	84 (66.1)	5 (41.7)	0.158

Neoplastic therapy including ICB therapy in female NSCLC patients; *n*, number of patients; %, percent. CTRL, patients receiving ICB without HVA therapy; COMB, patients receiving ICB with VA therapy; CTL-A4, cytotoxic T-lymphocyte antigen 4; PD-L1, programmed death ligand 1; PD-1, programmed cell death protein-1; ICB, immune checkpoint blockade.

**Table 5 ijms-26-03669-t005:** Application and combination forms of HVA therapy in addition to PD-1/PD-L1 inhibitor therapy in the COMB group, *n* = 61.

	Helixor^®^ A, i.v.	Helixor^®^ A, s.c.	Helixor^®^ M, i.v.	Helixor^®^ P, i.v.	Helixor^®^ P, s.c.	Helixor^®^ P, NA
**Mono, no combination, *n* (%)**	13 (21.3)	5 (8.2)	10 (16.4)	23 (37.7)	1 (1.6)	1 (1.6)
**Helixor^®^ A, s.c., *n* (%)**	5 (8.2)	0	0	0	0	0
**Helixor^®^ M, s.c., *n* (%)**	0	0	1 (1.6)	0	0	0
**Helixor^®^ P, s.c., *n* (%)**	0	0	0	1 (1.6)	0	0

Characterization of VA therapy; VA, *Viscum album* L.; NA, not applicable; *n*, number; %, percent; i.v., intravenous; s.c., subcutaneous.

**Table 6 ijms-26-03669-t006:** Median overall survival in NSCLC patients, *n* = 312.

	n	Events	Median [months]	95% CI [months]
NSCLC, CTRL	278	160	14.1	11.7–18.2
NSCLC, COMB	34	20	16.9	13.2–NA
Log rank test X^2^ = 1.8, *p* = 0.2				

NSCLC, non-small cell lung cancer; X^2^, chi-square; *p*, *p*-value; CI, confidence interval; CTRL, PD-1/PD-L1 inhibitors COMB, PD-1/PD-L1 inhibitors with HVA therapy.

**Table 7 ijms-26-03669-t007:** Median overall survival in female lung cancer patients, *n* = 137.

	n	Events	Median [months]	95% CI [months]
NSCLC female, CTRL	125	66	15.9	11.9–23.6
NSCLC female, COMB	12	4	NA	19.3–NA
Log rank test X^2^ = 3.6, *p* = 0.06				

NSCLC, non-small cell lung cancer; X^2^, chi-square; *p*, *p*-value; CI, confidence interval; CTRL, PD-1/PD-L1 inhibitors COMB, PD-1/PD-L1 inhibitors with HVA therapy.

## Data Availability

All relevant data are included in this manuscript.

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
