# Peer review of "Combined Immune Checkpoint Blockade and Helixor® Therapy in Oncology: Real-World Tolerability and Subgroup Survival (ESMO GROW)"

_ijms, 2025, doi:10.3390/ijms26083669_

Round 1
Reviewer 1 Report
Comments and Suggestions for Authors
This article investigates very interesting topic. In order to improve the manuscript I suggest:
- The abstract should be shorter. According to journal guidelines it should be maximum 200 words long. This one is app 500 words.
- In the section 2.1 study design the author should better describe used abbreviations for the groups (CTRL? and COMB?).
- In the section statistic, it should only be stated which statistical tests were used. Other information should be transferred to study design section.
- In the section results table 2 should be reorganized and the applied therapies should be listed by therapy line in which they were applied. Further the authors should add the information in which treatment line immunotherapy with or without HVA was applied for whole group. Further the authors should add the table with data about stage of diseases for NSCLC patients and therapy line in which immunotherapy with or without HVA was applied. Same data are needed for female patients with NSCLC. But it is unclear why the authors presented demographic data about whole group of patients and the survival analysis was done only for NSCLC patients. It is stated that the aim of the study was to evaluate the safety and efficacy of combining ICB with HVA therapy in a cohort of 405 oncological patients?! According to the presented results the title should also be modified.
- The section discussion should be improved .
- Limitations should also be improved.
Author Response
Point-by-point Response to Reviewer 1 Comments |
Dear Reviewer, thank you very much for taking the time to review this manuscript. Please find the detailed responses below and the corrections highlighted in track changes in the re-submitted files. In addition, please note, that we expanded the flowchart of the study (figure 1). Furthermore, the introduction was improved (page 2, line 69-92). Please see all other point-by-point responses to your comments below. With kind regards, The authors
|
Comments 1: This article investigates very interesting topic. In order to improve the manuscript I suggest: The abstract should be shorter. According to journal guidelines it should be maximum 200 words long. This one is app 500 words.
|
Response 1: Thank you for this comment. We have shortened the abstract according to your suggestions, and it contains now 192 words, please see page 1 lines 17-55.
|
Comments 2: In the section 2.1 study design the author should better describe used abbreviations for the groups (CTRL? and COMB?).
|
Response 2: Thank you for this point. We have improved section 2.1, study design, and also the description of the abbreviations CRTL and COMB in the study design, see pages 3-4, lines 119-161.
Comments 3: In the section statistic, it should only be stated which statistical tests were used. Other information should be transferred to study design section.
Response 3: Agree. We have therefore moved other information than statistical tests to the study design section, as suggested. Please see pages 4-5, lines 189-209.
Comments 4: In the section results table 2 should be reorganized and the applied therapies should be listed by therapy line in which they were applied. Further the authors should add the information in which treatment line immunotherapy with or without HVA was applied for whole group. Further the authors should add the table with data about stage of diseases for NSCLC patients and therapy line in which immunotherapy with or without HVA was applied. Same data are needed for female patients with NSCLC. But it is unclear why the authors presented demographic data about whole group of patients and the survival analysis was done only for NSCLC patients. It is stated that the aim of the study was to evaluate the safety and efficacy of combining ICB with HVA therapy in a cohort of 405 oncological patients?! According to the presented results the title should also be modified.
Response 4: Thank you for this point. We have reorganized Table 2 as suggested (see page 8, line 287). Additionally, as recommended, we have included two new sections: one on NSCLC patients (Chapter 3.3) and another specifically addressing female NSCLC patients (Chapter 3.4), both with information on their anti-neoplastic treatment as well as on the disease stage, see page 8, lines 292-310 and page 9, lines 312-332. We have also added two tables detailing therapy lines for immunotherapies with or without HVA—see Table 3 (page 8) and Table 4 (page 9).. Furthermore, we have revised the title as suggested. The terms "tolerability and effectiveness" have been removed, as effectiveness applies only to NSCLC patients. The revised title now reads: Combined Immune Checkpoint Blockade and Helixor® Viscum album Therapy in Oncological Patients: Real-World Evidence in Line with ESMO GROW Guidelines.” Comments 5: The section discussion should be improved
Response 5: Agree. We have, accordingly, modified the section discussion. Please see pages 13-14, lines 431-443 and lines 507-513.
Comments 6: Limitations should also be improved
Response 6: Thank you for this suggestion. We have, accordingly, improved the limitations, see page 15, lines 535-548.
|

Reviewer 2 Report
Comments and Suggestions for Authors
Reviewer Response
Manuscript Title: Tolerability and effectiveness of combined immune checkpoint 2 blockade and Helixor® Viscum album Therapy in Oncological 3 Patients: Real-World Evidence in line with ESMO GROW 4 Guidelines
General Comments: This manuscript is an interesting study examining the use of Real-world data (RDW) in assessing multimodal therapies in oncology. The study is well done with convincing data and suggests a potential therapeutic benefit of the complementary therapy viscum album extract alongside immune checkpoint blockades. There are a few areas in which clarification would be helpful as well as some minor editorial changes that should be made to enhance the manuscript’s quality. Please see below for detailed comments:
Minor comments:
- The main issue involves demographic composition and treatment protocols of the COMB group. Due to the small size and non-random assignment, there could be concerns about clinicians who provide complementary therapeutic options such as viscum album extract for patient types that don’t fully represent the patient population as a whole. For example, 17/21 patients enrolled in the COMB group had 0 or only 1 patient in specific tumor groups, also this group show a higher percentage of early vs late-stage disease. Furthermore, 80% of the ICB group has NSCLC but only 50% of the COMB group, whereas the COMB group had more breast cancer patients as well as being younger in general. Finally, are the dosages/compliance rates for viscum album extract as strict as they would be for clinical ICB therapies? If not were any of the deviations recorded? Further information about the demographics and treatment protocols of the COMB group could be added to the Discussion section to alleviate these concerns.
- Lines 220 and 221 states “A higher proportion of patients in the CTRL group received first-line ICB compared to CTRL and the difference was significant”. Is this incorrectly stated to leave out the COMB group? It also mentions that the difference was significant but I don’t see any of that data presented in the tables.
- Table 2 legend states “%, percent” but the % sign is not located in this table, assume that is represented by the numbers in parentheses but it is currently unclear. Table 3 and Supplementary Table 1 use a different system.
- Lines 242 and 243 mention patients that receive both intravenous and subcutaneous Helixor A but the same recognition isn’t given to those patients that receive both of Helixor M and Helixor P. Also, Table 2 appears to show one patient under the IV Helixor A column that is also in the IV Helixor A row. If this is supposed to mean something please explain more clearly.
- Lines 229 and 231-232 state “supplementary table” but do not refer to which specific supplementary table.
- Line 250 states “While 22 patients (6.4%) of patients”, please edit for clarity.
Author Response
|
Point-by-point Response to Reviewer 2 Comments |
Dear Reviewer, thank you very much for taking the time to review this manuscript. Please find the detailed responses below and the corrections highlighted in track changes in the re-submitted files. In addition, please note, that we expanded the flowchart of the study (figure 1). Furthermore, the introduction was improved (page 2, line 69-92). Additionally, as recommended by reviewer 1, we have included two new sections: one on NSCLC patients (Chapter 3.3) and another specifically addressing female NSCLC patients (Chapter 3.4), both with information on their anti-neoplastic treatment as well as on their disease stages, see page 8, lines 292-305 and page 9, lines 312-328. We have also added two tables detailing therapy lines for immunotherapies with or without HVA—see Table 3 (page 8) and Table 4 (page 9). Please see all other point-by-point responses to your comments below. With kind regards, The authors
|
|
|
|
|
Comments 1: General Comments: This manuscript is an interesting study examining the use of Real-world data (RDW) in assessing multimodal therapies in oncology. The study is well done with convincing data and suggests a potential therapeutic benefit of the complementary therapy viscum album extract alongside immune checkpoint blockades. There are a few areas in which clarification would be helpful as well as some minor editorial changes that should be made to enhance the manuscript’s quality. Please see below for detailed comments: |
|
|
Response 1: Thank you for the evaluation ,,that the study is well done with convincing data”.
Comments 2: Minor comments: The main issue involves demographic composition and treatment protocols of the COMB group. Due to the small size and non-random assignment, there could be concerns about clinicians who provide complementary therapeutic options such as viscum album extract for patient types that don’t fully represent the patient population as a whole. For example, 17/21 patients enrolled in the COMB group had 0 or only 1 patient in specific tumor groups, also this group show a higher percentage of early vs late-stage disease. Furthermore, 80% of the ICB group has NSCLC but only 50% of the COMB group, whereas the COMB group had more breast cancer patients as well as being younger in general. Finally, are the dosages/compliance rates for viscum album extract as strict as they would be for clinical ICB therapies? If not were any of the deviations recorded? Further information about the demographics and treatment protocols of the COMB group could be added to the Discussion section to alleviate these concerns.
Response 2: Thank you for this important point. We have discussed significant demographic differences between both groups in the discussion section, see pages 13-14, lines 431-442.In addition to concerns on sample size, non-random assignment, tumor distributions as well as dosage and compliance rates we have improved the ,limitations section, see page 15, lines 534-548.
Comments 3: Lines 220 and 221 states “A higher proportion of patients in the CTRL group received first-line ICB compared to CTRL and the difference was significant”. Is this incorrectly stated to leave out the COMB group? It also mentions that the difference was significant but I don’t see any of that data presented in the tables.
Response 3: Thank you for this important comment. We have changed this typo into the following sentence: “A higher proportion of patients in the CTRL group (64.8%) received first-line ICB compared to COMB (42.6%,), and the difference was significant. Here, the COMB group received 20.3% less first-line PD-1 inhibitor therapy compared to CTRL” In addition, we have included the missing row for first-line ICB in table 2. Please see page 7, lines 274-275 and table 2, last line.
Comments 4: Table 2 legend states “%, percent” but the % sign is not located in this table, assume that is represented by the numbers in parentheses but it is currently unclear. Table 3 and Supplementary Table 1 use a different system.
Response 4: Thank you for this hint. Indeed, the table 2 stated “%” in the legend but not in the table. We have now included the description “n (%)” in the table 2 which now utilizes the same system as table 3 (which is now table 5 on page 10) and supplementary table 1. Please see table 2 on page 8 and table 5 on page 10. Furthermore, we have added the description “n (%)” also in table 1 on page 7.
Comments 5: Lines 242 and 243 mention patients that receive both intravenous and subcutaneous Helixor A but the same recognition isn’t given to those patients that receive both of Helixor M and Helixor P. Also, Table 2 appears to show one patient under the IV Helixor A column that is also in the IV Helixor A row. If this is supposed to mean something please explain more clearly.
Response 5: Thank you for this comment. We have included now information on the combination of Helixor M iv with Helixor M sc as well as Helixor P iv wth Helixor P sc., see page 10, line 338-341. The patient in table 5 on page 10 receiving iv Helixor A was a patient that received Helixor A in two different time frames. However, as this is not a combination and rather the same medication, we have now deleted the line for Helixor A iv and included this patient in the line “mono, no combination” adding to 13 patients in total. We hope you agree.
Comments 6: Lines 229 and 231-232 state “supplementary table” but do not refer to which specific supplementary table.
Response 6: Agree. We have, accordingly, included a “1” to the supplementary table in order to specify it, see page 7 and 8, line 281, 284 and 286.
Comments 7: Line 250 states “While 22 patients (6.4%) of patients”, please edit for clarity.
Response 7: Thank you. We have edited this sentence accordingly: “In the CTRL group, 22 patients (6.4%) discontinued ICB therapy due to adverse events, whereas in the COMB group, only 3 patients (4.9%) did so.”, please see page 10, lines 348-349.

Round 2
Reviewer 1 Report
Comments and Suggestions for Authors
Comment 1: I still do not understand why you named the group who did not received HVA CTRL?
Comment 2: In the section results the captions of the tables 3 and 4 are not correct.
Comment 4: Line 211- A higher proportion of patients in the CTRL group (71.6%) received 17.7% more 211
first-line ICB compared to COMB (52.9%), and the difference was significant.
What 17.7% means?
Comment 5: it is unclear why the authors presented demographic data about whole group of patients and the survival analysis was done only for NSCLC patients. It is stated that the aim of the study was to evaluate the safety and efficacy of combining ICB with HVA therapy in a cohort of 405 oncological patients?! According to the presented results the title and the aim of the study still need to be modified.
Comment 6: In the discussion you should not repeat your results but you should compare them with the results of other studies.
Lines 373-381- you just mentioned some molecular pathways, but you did not perform that analysis. Thus you should move this part to the introduction, but with more details.
Lines 382-394- also should be moved to the introduction.
Same comment for the next paragraph.
In the discussion you should compare your results with the results of other studies!
Comment 7: The conclusion in the abstract is not in accordance with the results.
You cannot state suggest a potential gender-specific survival benefit since you did not perform analysis for male patients
Author Response
Point-by-point Response to Reviewer 1 Comments |
Dear Reviewer, thank you very much for taking the time to review this manuscript. Please find the detailed responses below and the corrections highlighted in track changes in the re-submitted files. Please see all other point-by-point responses to your comments below. With kind regards, The authors |
Comment 1: I still do not understand why you named the group who did not received HVA CTRL? Response 1: We designated the group receiving immune checkpoint inhibitors as the control (CTRL) group serving as the appropriate clinical benchmark against which we evaluated the potential additive or synergistic effects of Helixor Viscum album (HVA) therapy. The aim of our study was to assess whether the addition of HVA to an established therapeutic backbone (ICIs) could confer additional clinical benefits. Labeling the ICI-only group as “CTRL” therefore reflects its role as the reference treatment arm, consistent with conventions in clinical and translational research where the control group receives the current standard therapy. Comment 2: In the section results the captions of the tables 3 and 4 are not correct. Response 2: Thank you for pointing this out. We have reviewed and corrected the captions for Tables 3 and 4 to accurately reflect their contents. The updated captions are now consistent with the data presented and align with the text in the Results section. Comment 4: Line 211- A higher proportion of patients in the CTRL group (71.6%) received 17.7% more first-line ICB compared to COMB (52.9%), and the difference was significant. What 17.7% means? Response 4: Thank you for your comment. The 17.7% refers to the absolute difference in the proportion of patients receiving first-line immune checkpoint blockade (ICB) therapy between the CTRL and COMB groups. We have revised the sentence for clarity as follows: “A significantly higher proportion of patients in the CTRL group (71.6%) received first-line immune checkpoint blockade (ICB) therapy compared to the COMB group (52.9%), representing a 17.7% absolute difference between the groups, p=0.03", see line 211-223. Comment 5: it is unclear why the authors presented demographic data about whole group of patients and the survival analysis was done only for NSCLC patients. It is stated that the aim of the study was to evaluate the safety and efficacy of combining ICB with HVA therapy in a cohort of 405 oncological patients?! According to the presented results the title and the aim of the study still need to be modified. Response 5: We appreciate the reviewer’s comments and the opportunity to clarify this point. The primary objective of our study was to assess the tolerability/safety of the combined ICB and HVA therapy, which was evaluated in the entire cohort of 405 oncological patients, as stated in the manuscript. Accordingly, demographic data were presented for the whole group to support the tolerability analysis, which applies to all included patients regardless of tumor type. The secondary objective—to explore efficacy—was performed in a subgroup of patients with NSCLC, as this subgroup had adequate follow-up and sample size to allow for meaningful survival analysis. We believe this approach is methodologically sound and appropriately reflects the hierarchy of study objectives. However, in order that the title accurately reflect its design and focus, we changed it to the following: “Combined Immune Checkpoint Blockade and Helixor® Viscum album Therapy in Oncology: Real-World Tolerability and Subgroup Survival (ESMO GROW)”. Comment 6: In the discussion you should not repeat your results but you should compare them with the results of other studies. Response 6: We thank the reviewer for this observation. However, we respectfully disagree with the comment. In the Discussion section, we have made a conscious effort to avoid merely repeating our results. Where results are mentioned, they are presented in the context of comparison with existing literature, see in detail below, in order to highlight similarities and differences, and to place our findings into a broader clinical and scientific framework. Specifically, we have compared our tolerability data and survival outcomes with those reported in prior studies involving ICB and HVA therapy, and have discussed how our findings align with or diverge from these, see below: Lines 345-347: “The safety profile observed in this study, with fewer discontinuations of ICB therapy in the COMB group, aligns with literature highlighting the tolerability of add-on VA therapy when used alongside conventional cancer treatments (50, 51).” Comment: Lines 373-381- you just mentioned some molecular pathways, but you did not perform that analysis. Thus you should move this part to the introduction, but with more details. Comment: Lines 382-394- also should be moved to the introduction. Response: We respectfully disagree. The molecular mechanisms mentioned were not intended to represent novel findings from our own analyses but were included to contextualize and support our interpretation of potential gender-specific effects observed in our results. These mechanisms are discussed to underscore the plausibility of gender differences based on existing literature and biological pathways. Therefore, we believe it is appropriate to retain this discussion in the Discussion section, where we interpret our findings and integrate them with known mechanisms. In addition, it was made clear for every statement that these molecular pathways were discovered by other groups. Please see here: Line 389-396: “Several genes and molecular pathways have been implicated in gender-related differences in immune responses and PD-L1 expression such as estrogen-related, sex hormone-related, immune-related, oncogene and tumor suppression related molecular pathways (35, 36). Genetic alterations of these genes may influence the expression of PD-L1 and immune checkpoint blockade (35, 37). While gender-specific responses to both immunotherapy and complementary treatments have been explored in the literature (52, 53), more evidence is needed to elucidate whether female patients derive greater benefit from combinational approaches involving VA therapy.” Comment: Same comment for the next paragraph. Response: Thank you for your suggestion. We have moved the following text, as suggested, to the introduction, see line 56-62, as it rather describes the overall potential of VA therapy, which may not be directly connected to our findings: “Findings suggest the potential of VA therapy in improving patient prognosis across various settings (11, 29, 38, 39, 29). However, outcomes may be different between tumor types, i.e. immune-modulators such as PD-1 inhibitors or VA may be more effective in cancers like NSCLC because these tumors often have a higher mutational burden, producing more neo-antigens that can trigger an immune response (42). In contrast, pancreatic cancer tends to have a lower mutational burden and a dense, immunosuppressive TME, which may limit effectiveness of PD-1 inhibitors (43) and VA (44).” Comment: In the discussion you should compare your results with the results of other studies! Response: Thank you for your comment, which is the same as your comment 6. Please see our response 6. Comment 7: The conclusion in the abstract is not in accordance with the results. Response 7: Thank you for your comment. We have adapted the conclusion in the abstract and also in the manuscript text at the end. Please see abstract, lines 32-33 and in the manuscript text, lines 450-460. Comment You cannot state suggest a potential gender-specific survival benefit since you did not perform analysis for male patients Response: Thank you for your feedback. We respectfully disagree with this suggestion. In our analysis, we did not observe these changes in male patients. To support our statements, we have now included supplementary data. Additionally, we have updated the results section and the discussion section to include the following: 'This effect was not observed in male NSCLC patients, as shown in Supplementary Figure 2 and Supplementary Table 3.”, see lines 300-301 and line 378. |

Round 3
Reviewer 1 Report
Comments and Suggestions for Authors
The authors improved the manuscript, however I have one minor suggestion:
Comment 1: Lines 82-85- It is stated that the aim of the study was to evaluate the safety and efficacy of combining ICB with HVA therapy in a cohort of 405 oncological patients?! Please clarify that the safety was investigated in the whole group and the efficacy in 312 NSCLC patients.
Author Response
Point-by-point Response to Reviewer 1 Comments |
Dear Reviewer, thank you very much for taking the time to review this manuscript. Please find our response below and the corrections highlighted in track changes in the re-submitted file (word file). With kind regards, The authors |
Comment 1: Lines 82-85- It is stated that the aim of the study was to evaluate the safety and efficacy of combining ICB with HVA therapy in a cohort of 405 oncological patients?! Please clarify that the safety was investigated in the whole group and the efficacy in 312 NSCLC patients. Response 1: Thank you for your helpful comment. We have revised the sentence accordingly and it reads now: “Our study utilized real-world registry data to evaluate the safety of combining ICB with HVA therapy in a cohort of 405 oncological patients and the efficacy of this combination in a subgroup of 312 patients with non-small cell lung cancer (NSCLC).” Please refer to lines 80-81 in the marked-up text (word file). |
